# REINFORCING QUERY-LEVEL META-AGENTS

## ABSTRACT

This paper proposes a query-level meta-agent named FLOWREASONER to automate the design of query-level multi-agent systems, i.e., *one system per user query*. Our core idea is to incentivize a reasoning-based meta-agent via external execution feedback. Concretely, by distilling DeepSeek R1, we first endow the basic reasoning ability regarding the generation of multi-agent systems to FLOWREASONER. Then, we further enhance it via reinforcement learning (RL) with external execution feedback. A multi-purpose reward is designed to guide the RL training from aspects of performance, complexity, and efficiency. In this manner, FLOWREASONER is enabled to generate a personalized multi-agent system for each user query via deliberative reasoning. Experiments on both engineering and competition code benchmarks demonstrate the superiority of FLOWREASONER. Remarkably, it surpasses o1-mini by $10.52\%$ accuracy across three benchmarks. All the code is included in the Supplemental Material.

## 1 INTRODUCTION

Large language models (LLMs) (Achiam et al., 2023; Reid et al., 2024; Team, 2024a; Yang et al., 2024; Liu et al., 2024) have exhibited remarkable power in various meaningful yet challenging domains, like chatbots (OpenAI, 2022), code (CognitionAI, 2024), math (OpenAI, 2024), robotics (Kim et al., 2024), etc. LLM-based multi-agent systems (Hong et al., 2023; Wu et al., 2023; Li et al., 2023), which are characterized by planning, reasoning, tool use, and memory, become the foundation of these LLM-driven applications.[1] While effective, most of them are manually designed, increasing human resource costs and limiting scalability.

To mitigate this challenge, early automatic methods are proposed to optimize the prompts (Yuksekgonul et al., 2024; Khattab et al., 2024; Zhou et al., 2024a; Yang et al., 2023) or hyperparameters (Saad-Falcon et al., 2024). But they still rely on the fixed workflow of the multi-agent system, which requires human effort to manually design workflows for each new scenario. From this motivation, various graph-based methods (Zhuge et al., 2024; Liu et al., 2023; Zhang et al., 2024a; Feng et al., 2025) formulate the workflows as graphs or networks and automate the workflow designs. However, the structural complexity of graphs limits their scalability (Hu et al., 2024). To overcome this limitation, state-of-the-art methods represent the multi-agent systems as programming codes (Hu et al., 2024) and prompt a performant LLM, e.g., GPT-4o, as a *meta-agent* to optimize workflows via complex search algorithms on carefully designed search sets (Zhang et al., 2024b; Shang et al., 2024; Zhang et al., 2025).

These previous methods focus on **task-level meta-agents**, generating merely a single **task-specific multi-agent system** that applies to one kind of task, e.g., *code generation task*, as in Figure 1 (a). However, for individual user queries, these one-size-fits-all systems lack the capability for automatic adaptation. To enhance the adaptability of multi-agent systems for individual user queries, this paper aims to design a **query-level meta-agent** to generate a **query-specific multi-agent system** for each user query, e.g., *build a 2048 game*, as shown in Figure 1 (b).

We first identify that the success of task-level meta-agents largely depends on carefully designed search sets, as they rely on complex search algorithms. However, such search sets are unavailable in the setting of query-specific multi-agent systems. To address this issue, instead of relying on search algorithms, we propose to integrate **external execution feedback** of the generated multi-agent system, based on which a **reasoning-driven meta-agent** is leveraged to polish the system.

---

[1]This paper defines a *multi-agent system* as a system consisting of multiple agents operating under a *workflow*.

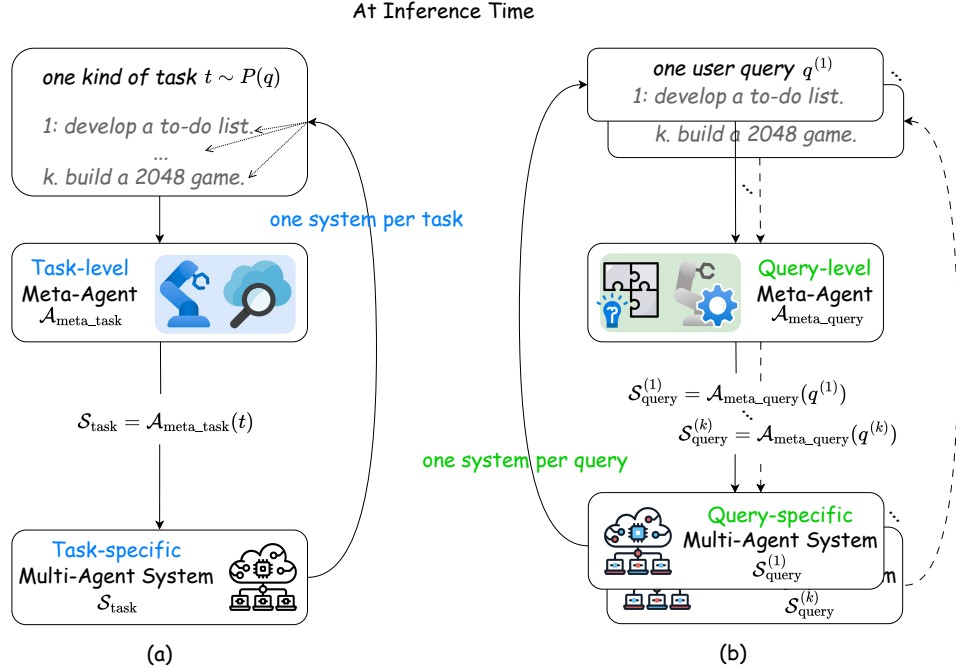

Figure 1: **Task-Level Meta-Agents vs. Query-Level Meta-Agents at Inference Time.** $q$ denotes a user query, e.g., *build a 2048 game*. $t \sim P(q)$ denotes one kind of task, e.g., *code generation task*, which is a distribution of user queries. Given $t$, previous task-level meta-agent $\mathcal{A}_{\text{meta\_task}}$ aims to search a task-specific multi-agent system $\mathcal{S}_{\text{task}}$ to solve all queries sampled from $t$, i.e., one system per task. Differently, given one user query $q^{(i)}$, our query-level meta-agent $\mathcal{A}_{\text{meta\_query}}$ conducts reasoning and output a query-specific multi-agent system $\mathcal{S}_{\text{query}}^{(i)}$ for $q^{(i)}$, i.e., one system per query.

We dub such a meta-agent as FLOWREASONER. We first synthesize thousands of warm-up SFT data using DeepSeek R1-671B (Team, 2025a) as the meta-agent to generate multi-agent systems and process user queries individually. These synthetic data are then used to finetune DeepSeek-R1-Distill-Qwen-7B, enabling basic reasoning for multi-agent system generation. Furthermore, we enhance its reasoning capabilities for generating novel query-level multi-agent systems through reinforcement learning (RL), incorporating external execution feedback. A **multi-purpose reward** is designed to guide RL training, focusing on *performance*, *complexity*, and *efficiency*. During inference, FLOWREASONER leverages deliberative reasoning to generate a novel query-level multi-agent system for each user query, achieving *one system per user query*. The main contributions are summarized as follows:

- We propose a query-level meta-agent termed FLOWREASONER to automate the designs of query-level multi-agent systems, improving their adaptability in real-world scenarios.

- We train FLOWREASONER to reason from external execution feedback via RL, guided by a multi-purpose reward considering performance, complexity, and efficiency.

- We demonstrate the superiority of FLOWREASONER via extensive experiments and open-source it.

## 2 RELATED WORK

**LLM-Based Multi-Agent Systems.** LLM-based multi-agent systems (Guo et al., 2024; Du et al., 2023; Chen et al., 2023b; Park et al., 2023; Huang et al., 2024; Jin et al., 2025) serve as the foundation of various LLM-powered real-world applications, e.g., code intelligence (CognitionAI, 2024), computer use (Anthropic, 2024), and deep research (OpenAI, 2025). LLM-based agents are equipped with planning capabilities, database access, and tool function invocation. These agents collaborate within a multi-agent system, leading to promising performance. Most multi-agent systems are manually designed, increasing the costs of human resources and limiting the scalability. To address this issue, researchers propose automation methods to automate the design of multi-agent

systems. Early methods (Yang et al., 2023; Khattab et al., 2024; Zhou et al., 2024a) are proposed to automate the agentic designs via optimizing prompts (Yuksekgonul et al., 2024) or hyper-parameters (Saad-Falcon et al., 2024). For example, (Chen et al., 2023a; Yuan et al., 2024; Fernando et al., 2023) adopt the evolution algorithms to automate the agent profiling.

Although effective, they merely optimize the agents but keep the workflow of the multi-agent system fixed, which still requires human effort to manually design for each new scenario. To address this problem, several methods (Li et al., 2024; Zhou et al., 2024b; Zhang et al., 2024c; Shang et al., 2024; Zhuge et al., 2023) are proposed to automate the design of the entire agentic workflow. For example, researchers (Zhuge et al., 2024; Khattab et al., 2024; Liu et al., 2023; Zhang et al., 2024a; Feng et al., 2025) formulate the workflows as graphs or networks and then optimize the connections between nodes. To improve the efficiency, ADAS (Hu et al., 2024) proposes to use programming codes to represent both agents and workflows. It also introduces a meta-agent to generate these workflows and presents the meta-agent search to optimize the designs of multi-agent systems. At the same time, AFLOW (Zhang et al., 2024b) also adopts the code representation but proposes Monte Carlo Tree Search (MCTS) to optimize it. In addition, MaAS (Zhang et al., 2025) presents the agentic supernet and then conducts the multi-agent architecture search. Differently, ScoreFlow (Wang et al., 2025) trains a workflow generator to generate better workflows via direct preference optimization (DPO) (Rafailov et al., 2023).

This paper argues that the previous methods are task-level meta-agents. As shown in Figure 1 (a), they merely generate a single task-specific multi-agent system for solving one kind of task. However, these one-size-fits-all systems are rigid and unable to automatically adapt or customize to individual user queries within a task. From this motivation, we aim to propose a query-level meta-agent to generate a query-specific multi-agent system for each user query as shown in Figure 1 (b).

**Reasoning in LLMs.** The ability to reason is essential for LLMs, enabling them to emulate human thinking patterns. Pioneering work (Wei et al., 2022; Kojima et al., 2022) has facilitated this by prompting LLMs to think step by step. Beyond this approach, reasoning capabilities are further enhanced through frameworks such as self-correction (Kumar et al., 2024), self-critique (Ke et al., 2023), debate (Liang et al., 2023; Du et al., 2023), and plan-and-solve (Wang et al., 2023). Additionally, efforts like (Hao et al., 2024; Goyal et al., 2023) seek to transition LLMs' reasoning processes into the latent space. OpenAI has advanced reasoning in LLMs by developing the o1 model, demonstrating the potential for improvement through test-time scaling. Inspired by this, models such as QwQ (Team, 2024c), QvQ (Team, 2024b), DeepSeek (Team, 2025a), and Kimi (Team, 2025b) have followed suit, developing o1-like reasoning architectures. Moreover, OpenAI's o3 model has been announced to achieve promising results on the ARG-AGI benchmark (ARC-AGI, 2024). LLMs progressively shift from intuitive processing (System 1) to deliberative reasoning (System 2) (Li et al., 2025). Besides, researchers demonstrate that reasoning can improve safety (Liu et al., 2025a) and alleviate hallucination (Gao et al., 2025). However, Chen et al. (2024) examines the overthinking problem observed in o1-like models. To alleviate this problem, token efficiency methods (Liu et al., 2025b) are proposed to reduce the token costs while maintaining the reasoning quality. This paper develops an o1-like reasoning model to serve as a query-level meta-agent, getting rid of complex search algorithms and the carefully designed search set.

## 3 PROBLEM DEFINITION

We denote a user **query** as $q$, e.g., *build a 2048 game*. Then a user **task** is defined as a distribution of user queries, denoted as $t = P(q)$, e.g., *code generation task*. A multi-agent system is denoted as $S = \{A, W\}$, where $A = \{A_1, ..., A_n\}$ denotes the agents in the system and $W$ is the *workflow* of collaboration among the agents.

As shown in Figure 2 (a), in traditional multi-agent systems (Hong et al., 2023; Wu et al., 2023; Li et al., 2023), the agents and the workflows are designed manually according to one kind of task $t = P(q)$, as formulated $S_{task} = \mathcal{H}(t)$, where $\mathcal{H}$ denotes human experts, and $S_{task}$ denotes the **task-level multi-agent system** which is fixed for all queries in one task. This kind of manually designed multi-agent system leads to extensive human costs. Besides, such a one-size-fits-all system fails to allocate inference resources dynamically for different user queries within the task.

As illustrated in Figure 2 (b), search-based automatic multi-agent systems (Zhang et al., 2024b; Hu et al., 2024; Chen et al., 2023a) are proposed to reduce human effort. Specifically, these approaches

Figure 2: **Architectural Comparison of Three Multi-Agent Systems**. (a) Manually-designed Multi-agent System, (b) Search-based Automatic Multi-agent System, and (c) Reasoning-based Automatic Multi-agent System.

first prompt an LLM, such as GPT-4o, to act as a meta-agent, generating multiple candidate multi-agent system designs. Subsequently, complex search algorithms are employed on the carefully designed search set to identify the optimal system for completing the task. We denote this kind of meta-agent as **task-level meta-agent** $\mathcal{A}_{\text{meta\_task}}$ since they can merely generate one **task-level multi-agent system** $\mathcal{S}_{\text{task}}$ to solve one kind of task $t$. Although effective, they are still one-size-fits-all systems. Besides, the search algorithm is time-consuming and relies on the search set, which is absent in one user query.

## 4 META-AGENT FLOWREASONER

To solve these problems, we develop a reasoning-based automatic multi-agent system shown in Figure 2 (c). Concretely, by guiding the model to reason from external execution feedback, we train a **query-level meta-agent** denoted as $\mathcal{A}_{\text{meta\_query}}$, which can automatically propose a novel **query-level multi-agent system** for each user query $q$, as formulated $\mathcal{S}_{\text{query}} = \mathcal{A}_{\text{meta\_query}}(q)$. Then, $\mathcal{S}_{\text{query}}$ accomplishes the specific user query and obtain the result $a$, i.e., $a = \mathcal{S}_{\text{query}}(q)$. Subsequently, the evaluator $\mathcal{E}$ evaluates the performance of the proposed multi-agents system $\mathcal{S}_{\text{query}}$ by comparing the result $a$ with ground truth $a_{\text{gt}}$ as formulated $\mathcal{E}(a, a_{\text{gt}})$. Our proposed method is more practical

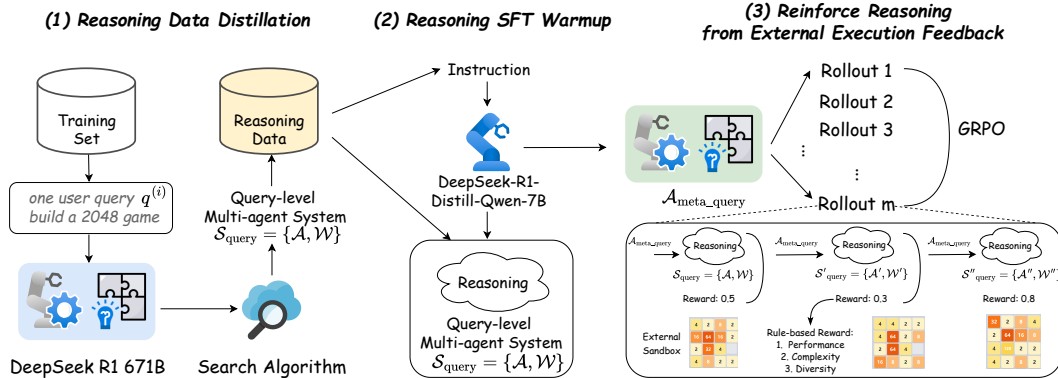

Figure 3: **Training Pipeline of FLOWREASONER.** It consists of (1) Reasoning Data Distillation, (2) Reasoning SFT Warmup, (3) Reinforce Reasoning from external execution feedback.

in real-world scenarios, as it can design an optimal multi-agent system for each specific query. The training pipeline of our method is demonstrated in Figure 3.

## 4.1 LEARN TO REASON

**Reasoning Data Syntheses.** To enable our model to learn how to reason workflows based on external execution feedback, we first generate multi-round reasoning data using R1-671B(Team, 2025a). For a given input query $q$, the R1 model generates $l$ rounds of reasoning $\mathcal{R}_{\text{query}}$ and a multi-agent system $\mathcal{S}_{\text{query}}$ with external execution feedback at each round as follows:

$$\{\{\mathcal{R}_{\text{query}}, \mathcal{S}_{\text{query}}\}^{(1)}, ..., \{\mathcal{R}_{\text{query}}, \mathcal{S}_{\text{query}}\}^{(l)}\} = \mathcal{A}_{\text{meta\_query}}(q). \tag{1}$$

Then, we concatenate the model's $l$ rounds of reasoning and multi-agent system to form a reasoning process $\hat{\mathcal{R}}$ and final multi-agent system $\hat{\mathcal{S}}$, and pair it with the instruction $\mathcal{I}$ and query $q$ to construct our training sample $\{\mathcal{I}, q, \hat{\mathcal{R}}, \hat{\mathcal{S}}\}$. Based on this, we construct a warmup SFT dataset $\mathcal{D}$.

**Reasoning SFT Warmup.** After creating the reasoning training dataset $\mathcal{D}$, we proceed to perform reasoning SFT for warmup. We input the instruction $\mathcal{I}$ and query $q$, then guide DeepSeek-R1-Distill-Qwen-7B to output reasoning process $\hat{\mathcal{R}}$ and final multi-agent system $\hat{\mathcal{S}}$. It can be formulated as follows:

$$\mathcal{L}_{\text{SFT}} = -\mathbb{E}_{(\mathcal{I}, q, \hat{\mathcal{R}}, \hat{\mathcal{S}}) \sim \mathcal{D}} \log P_\theta(\hat{\mathcal{R}}, \hat{\mathcal{S}} \mid \mathcal{I}, q), \tag{2}$$

where $\theta$ denotes the model parameters. Through SFT, we unlock the model's reasoning ability regarding workflow generation.

## 4.2 REINFORCE REASONING FROM EXTERNAL EXECUTION FEEDBACK

After the SFT stage, we use reinforcement learning phase to further enhance the model's reasoning capabilities through workflows built on external execution feedback. We aims to leverage feedback to improve performance on complex tasks in multi-round multi-agent systems. Following the DeepSeek-R1-Zero (Team, 2025a), we adopt standard GRPO (Grouped Relative Policy Optimization) as our training method. GRPO works by sampling multiple outputs for each query and computing relative advantages based on the rewards these outputs receive. The policy is then updated by maximizing an objective function incorporating these relative advantages. The GRPO objective function can be expressed as follows:

$$\mathcal{L}_{\text{GRPO}}(\theta) = \mathbb{E}_{q \sim t, \{o_i\}_{i=1}^{G} \sim \pi_{\theta_{\text{old}}}(O|q)}$$

$$\frac{1}{G} \sum_{i=1}^{G} \left\{ \min\left[r_{\text{ratio}}, \text{clip}(r_{\text{ratio}}, 1 - \varepsilon, 1 + \varepsilon)\right] \cdot \hat{E}_{i,m} - \beta \mathbb{D}_{\text{KL}}(\pi_\theta \| \pi_{\text{ref}}) \right\}, \tag{3}$$

where $G$ represents the number of sampled trajectories in each group, $o_i$ denotes the $i$-th trajectory, $|o_i|$ is the length of trajectory $o_i$, $r_{\text{ratio}} = \frac{\pi_\theta(o_{i,m}|q,o_{i,<m})}{\pi_{\theta_{\text{old}}}(o_{i,m}|q,o_{i,<m})}$ represents the probability ratio between

new and old policies, $\hat{E}_{i,m}$ is the estimated advantage for the $m$-th token in the $i$-th trajectory, $\varepsilon$ is a small positive clipping parameter that prevents excessively large update steps, and the KL regularization term constrains policy drift to maintain stability during training.

Since we can obtain a reward from the external environment in every round, we use process reward supervision. The reward for each reasoning round is normalized, and the advantage function is calculated as the sum of subsequent normalized rewards.

$$\hat{E}_{i,m} = \sum_{j \geq T} \tilde{r}_j^{(i)}. \tag{4}$$

Here, the normalized reward for the $j$-th round of the $i$-th candidate output is defined as $\tilde{r}_j^{(i)} = k^j \cdot \frac{r_j^{(i)} - \text{mean}(\mathbf{R})}{\text{std}(\mathbf{R})}$, where $k$ is a scaling factor and $T$ is a threshold used to exclude the first $T$ items from the calculation. The set $\mathbf{R}$ represents the list of scores for each round across all candidates, and $r_j^{(i)}$ is the score of candidate $i$ in round $j$. Each score $r_j^{(i)}$ is calculated by the performance of the proposed solution (i.e., pass rate), algorithm complexity (i.e., complexity score of abstract syntax tree) and diversity (i.e., distinctness ratio followed by Chen et al. (2024)) of workflow.

### 4.3 GENERATE MULTI-AGENT SYSTEMS WITH FLOWREASONER

Constructing a multi-agent system is essentially an optimization problem with the goal of designing an optimized system, $\mathcal{S}_{\text{query}}^*$, that responds to user queries. When a user submits a query $q$, the system produces an answer $a = \mathcal{S}_{\text{query}}(q)$. The performance of the system in each round is evaluated using an external feedback function $\mathcal{E}(a, a_{\text{gt}})$, where $a_{\text{gt}}$ represents the ground truth answer. In this framework, a meta-agent, $\mathcal{A}_{\text{meta\_query}}$, is responsible for optimizing the workflow with external execution feedback $\mathcal{E}(a, a_{\text{gt}})$ in every round. The pass rate of the proposed solution for the given query, as the key performance indicator of the system, serves as the external execution feedback. The optimization space is defined by all possible configurations of nodes and edges. Here, nodes represent various parameters (such as language models, prompts, temperature, and output formats), and edges capture the interactions or data flows between these nodes.

By representing both nodes and edges in code and employing predefined operators (such as Ensemble, Review, and Revise) along with a custom operator to combine these elements, FLOWREASONER utilizes an $l$-round optimization process same as Aflow (Zhang et al., 2024b) to arrive at the final multi-agent system:

$$\mathcal{S}_{\text{query}}^* = \arg \max_{\mathcal{S}_{\text{query}} \in \mathbf{S}} \mathcal{E}(\mathcal{S}_{\text{query}}(q), a_{\text{gt}}), \tag{5}$$

where $\mathcal{S}_{\text{query}}^*$ is the optimal multi-agent system refined through optimization. By optimizing a multi-agent system through iterative external execution feedback, FLOWREASONER can construct a highly adaptive system that maximizes the accuracy and performance of solution for a query.

## 5 EXPERIMENTS

**Datasets.** Given our focus on creating workflows tailored to individual user queries rather than the general task, we restrict our scope to *code generation tasks*, as they can provide test cases as external execution feedback for the workflow construction process. Among various benchmarks, we consider three representative datasets: BigCodeBench (Zhuo et al., 2024), which emphasizes engineering-oriented tasks, and two algorithmically focused benchmarks, HumanEval (Chen et al., 2021) and MBPP (Austin et al., 2021). This selection enables us to comprehensively evaluate workflow discovery across a diverse spectrum of code generation challenges.

**Baselines.** In line with prior work (Hu et al., 2024; Zhang et al., 2024b), we evaluate FLOWREA-SONER against three categories of baselines: (1) single-model direct invocation, where a single LLM is prompted to solve the problem without additional structure; (2) manually designed workflows, including Self-Refine (Madaan et al., 2023), LLM-Debate (Du et al., 2023), and LLM-Blender (Jiang et al., 2023), which incorporate human-crafted reasoning strategies; and (3) automated workflow optimization methods, such as Aflow (Zhang et al., 2024b), ADAS (Hu et al., 2024), and MaAS (Zhang et al., 2025), which construct workflows through search or optimization over possible reasoning

Table 1: **Performance Evaluation.** Accuracy comparison across three code benchmarks for three categories of baselines - individual models, manual workflows, and automated workflow methods - alongside our FLOWREASONER-14B. For manual methods, model names in parentheses indicate the worker model used.

| Type | Method | BigCodeBench | HumanEval | MBPP | Overall |
|---|---|---|---|---|---|
| Vanilla | o1-mini | 57.67 | 95.42 | 74.19 | 71.37 |
| | GPT-4o-mini | 56.33 | 88.55 | 71.73 | 68.60 |
| Manual | Self-Refine (4o-mini) (Madaan et al., 2023) | 54.78 | 89.83 | 69.64 | 67.29 |
| | LLM-Debate (4o-mini) (Du et al., 2023) | 56.88 | 91.64 | 70.28 | 68.69 |
| | LLM-Blender (4o-mini) (Jiang et al., 2023) | 57.46 | 89.44 | 76.39 | 71.25 |
| | Self-Refine (o1-mini) (Madaan et al., 2023) | 56.68 | 94.74 | 73.64 | 70.63 |
| | LLM-Debate (o1-mini) (Du et al., 2023) | 57.25 | 95.83 | 74.28 | 71.33 |
| | LLM-Blender (o1-mini) (Jiang et al., 2023) | 59.51 | 96.37 | 78.65 | 74.22 |
| Auto | AutoAgents (Chen et al., 2023a) | 56.65 | 88.91 | 72.03 | 68.92 |
| | ADAS (Hu et al., 2024) | 53.87 | 84.26 | 68.47 | 65.48 |
| | Aflow (Zhang et al., 2024b) | 59.83 | 94.15 | 82.40 | 75.63 |
| | MaAS (Zhang et al., 2025) | 60.33 | 95.42 | 84.16 | 76.81 |
| | ScoreFlow (Wang et al., 2025) | 60.71 | 95.67 | 84.73 | 77.25 |
| Ours | FLOWREASONER-14B | **63.53** | **97.26** | **92.15** | **81.89** |

Table 2: **Ablation Study on Model Sizes and Training Stages.** Accuracy(%) comparison across three code benchmarks for models of different sizes (7B/14B) at both the Supervised Fine-Tuning (SFT) stage and the combined SFT with Reinforcement Learning (SFT+RL) stage.

| Stage | Size | BigCodeBench | HumanEval | MBPP | Overall |
|---|---|---|---|---|---|
| SFT | 7B | 61.79 | 96.38 | 87.22 | 78.89 |
| SFT+RL | 7B | 62.78 | 96.95 | 89.86 | 80.53 |
| SFT | 14B | 62.83 | 97.18 | 91.91 | 81.50 |
| SFT+RL | 14B | 63.53 | 97.26 | 92.15 | 81.89 |

structures. This comparison enables a comprehensive assessment of FLOWREASONER's effectiveness relative to both static and adaptive baselines.

**Implementation Details.** For the manually designed workflow baselines, we employed both o1-mini and GPT-4o-mini as worker models for each method. For the automated workflow optimization baselines, we adopted the original configurations as described in MaAS (Zhang et al., 2025). In our proposed method, FLOWREASONER, we trained two variants of DeepSeek-R1-Distill-Qwen with 7B and 14B parameters, respectively, and used o1-mini as the worker model. We fixed the number of workflow iterations to 10. To assess performance, we used the standard pass@1 metric for code accuracy, consistent with prior work (Chen et al., 2021). More details can be found in Appendix A.

## 5.1 EXPERIMENT RESULTS

**Performance Comparison.** Table 1 presents the performance comparison between our proposed method and the baselines. FLOWREASONER-14B consistently outperforms all competing approaches across the three benchmark datasets. Notably, it achieves an overall improvement of 5 percentage points over the strongest baseline, MaAS, and exceeds the performance of its underlying worker model, o1-mini, by a substantial margin of 10%. These results highlight the effectiveness of our workflow-based reasoning framework in enhancing code generation accuracy.

**Ablation on Model Size and Training Stages.** To investigate the impact of model size and training stages on performance, we conducted an ablation study comparing different configurations of FLOWREASONER in Table 2. We observed that the 14B variant consistently outperformed the 7B counterpart across all benchmarks, indicating a positive correlation between model scale and reasoning effectiveness. Furthermore, within each model size, versions trained with both SFT and RL exhibited notable improvements over those trained with SFT alone, demonstrating the complementary benefits of incorporating RL in enhancing workflow-guided code generation.

**Ablation of Meta-agents and Workers.** To analyze the impact of meta-agent and worker selection, we conducted an ablation study on the BigCodeBench dataset. Figure 4 presents performance

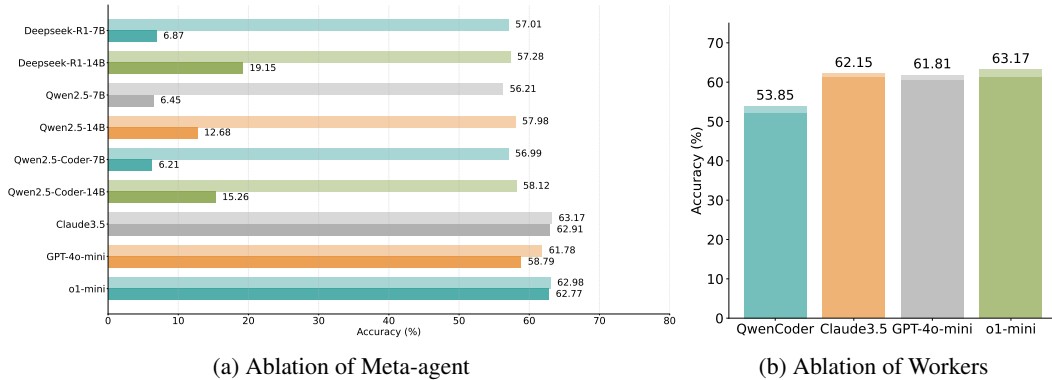

(a) Ablation of Meta-agent        (b) Ablation of Workers

Figure 4: **Ablation of Meta-agent and Workers.** (a) Accuracy of different meta-agents with o1-mini as workers. (B) Accuracy of the generated workflow with different workers. The light bars show results with the initial workflow (multi-round iterations), while the dark bars show results without it.

Table 3: **Generalization Evaluation.** Accuracy of our trained meta-agent FLOWREASONER-7B/14B when paired with alternative workers including Qwen2.5 Coder, Claude, and GPT4o-mini.

| Worker/Meta-agent | FLOWREASONER-7B (SFT+RL) | | | FLOWREASONER-14B (SFT+RL) | | |
|---|---|---|---|---|---|---|
| | BigCodeBench | HumanEval | MBPP | BigCodeBench | HumanEval | MBPP |
| **o1-mini** | 62.77 | 96.95 | 89.86 | 63.53 | 97.26 | 92.15 |
| **Qwen2.5Coder-32b** | 50.17 | 92.89 | 80.40 | 52.67 | 93.69 | 78.90 |
| **Claude3.5** | 60.67 | 96.07 | 87.63 | 61.12 | 96.52 | 89.82 |
| **GPT-4o-mini** | 59.18 | 94.24 | 82.19 | 59.75 | 94.52 | 82.27 |

comparisons under various meta-agent and worker configurations. As shown in Figure 4 (a), open-source models exhibited poor performance when paired with o1-mini as the worker and no initial workflow, frequently generating error-prone workflows. This highlights a limitation of current open-source models, which struggle to produce reliable workflows solely based on instruction prompts and are heavily reliant on predefined, manually crafted workflows. In contrast, API-based models demonstrated stronger performance, likely attributable to their superior instruction-following capabilities. Figure 4 (b) further examines worker effectiveness when using a high-performing meta-agent (Claude 3.5). We compared the open-source model Qwen2.5-Coder-32B and three API-based models, including Claude 3.5, GPT-4o-mini, and o1-mini. Among these, o1-mini achieved the best overall performance, suggesting its suitability as a worker model in the FLOWREASONER framework.

**Generalization Capability.** To evaluate whether FLOWREASONER, trained with o1-mini as the worker, can generalize its planning capabilities to alternative workers, we conducted experiments by substituting the worker with Qwen2.5-Coder, Claude, and GPT-4o-mini, while keeping the meta-agent fixed as either FLOWREASONER-7B or FLOWREASONER-14B. As shown in Table 3, FLOWREASONER exhibits a notable degree of transferability, maintaining consistent performance across different worker models on the same tasks. These results suggest that the planner is not tightly bound to a specific worker and can adapt its strategies effectively across diverse execution agents.

**Case Study.** Figure 5 illustrates two example workflows generated by FLOWREASONER-14B for representative tasks from BigCodeBench and HumanEval, respectively. The workflow corresponding to the BigCodeBench task exhibits greater complexity, reflecting the more challenging and engineering-oriented nature of the task. In contrast, the HumanEval workflow is substantially more concise, aligning with the relative simplicity and algorithmic focus of the task. These examples demonstrate FLOWREASONER 's ability to adapt the structure and granularity of workflows based on task complexity. More cases can be found in Appendix B.

## 6 CONCLUSION

In this paper, we present FLOWREASONER, a query-level meta-agent designed to automate the creation of personalized multi-agent systems for individual user queries. Unlike previous task-level approaches that create fixed, one-size-fits-all systems, FLOWREASONER dynamically generates tai-

```python
async def __call__(self, problem: str, test_cases: list):
    max_retries = 3
    solutions = []

    for _ in range(max_retries):
        for attempt in range(3):
        # Try generating code up to 3 times
            solution = await self.custom_code_generate(
                problem=problem, instruction=prompt_custom.GENERATE_CODE_PROMPT
                )
            test_result = await self.test(
                problem=problem, solution=solution['response'], test_cases=test_cases
            )

            if test_result['result']:
                solutions.append(solution['response'])
                break  # Break the inner loop if a correct solution is found
            elif attempt < 2:
            # If it's not the last attempt, try to improve the solution
                error_analysis = await self.custom(
                input=f"Problem: {problem}\nFailed solution: {solution['response']}\
                    nError: {test_result['solution']}", instruction=prompt_custom.
                    ERROR_ANALYSIS_PROMPT
                )
                prompt_custom.GENERATE_CODE_PROMPT = f"{prompt_custom.
                    GENERATE_CODE_PROMPT}\n{error_analysis['response']}"

        if not solutions:  # If no correct solution was found after 3 attempts
            solutions.append(solution['response'])  # Add the last generated solution

    best_solution = await self.sc_ensemble(solutions=solutions, problem=problem)
    return best_solution['response'], self.llm.cost_manager.total_cost
```

(a) Workflow generated by FLOWREASONER-14B for the BigCodeBench task of *generating traffic data for various vehicle types over a specified number of hours, saving the data to a CSV file with columns, and plotting it in a line chart.*

```python
async def __call__(self, problem: str, test_cases: list):
    solution = await self.custom_code_generate(problem=problem, entry_point=
        entry_point, instruction=prompt_custom.CODE_GENERATE_PROMPT)

    # Add review to check the solution
    review = await self.custom(
    input=f"Problem: {problem}\nGenerated solution: {solution['response']}",
        instruction=prompt_custom.REVIEW_PROMPT
    )

    # Improve the solution based on the review information
    improved_solution = await self.custom_code_generate(
    problem=problem, instruction=f"{prompt_custom.CODE_IMPROVE_PROMPT}\n\nConsider
        this review: {review['response']}"
    )

    return improved_solution['response'], self.llm.cost_manager.total_cost
```

(b) Workflow generated by FLOWREASONER-14B for the HumanEval task of *splitting the string into words and returning an array of the words.*

Figure 5: Cases of Workflows generated by FLOWREASONER-14B for the tasks of BigCodeBench and HumanEval.

lored workflows for each specific query through reasoning-based optimization. The system leverages external execution feedback and reinforcement learning with multi-purpose rewards focusing on performance, complexity, and efficiency to generate optimized workflows without relying on complex search algorithms or carefully designed search sets. Experimental results demonstrate that FLOWREA-SONER-14B outperforms both manually designed workflows and existing automated methods across multiple code generation benchmarks, notably improving o1-mini's performance by overall 10.52% on three benchmarks, thus proving the effectiveness and adaptability of reasoning-driven workflow generation. Besides, the observation when pairing FLOWREASONER with different worker models further confirms its generalization capabilities. Our approach reduces human resource costs while enhancing scalability by enabling more adaptive and efficient multi-agent systems that dynamically optimize their structure based on specific user queries rather than relying on fixed workflows for entire task categories.

## ETHICS STATEMENT

Our research on automated multi-agent system design does not involve human subjects and does not raise concerns regarding discrimination, bias, fairness, privacy, or security. The entire construction process focuses solely on technical methodology for code generation workflow optimization without involving sensitive data or human participation.

## REPRODUCIBILITY STATEMENT

To ensure the reproducibility of our research, we have included all the code required to reproduce our experimental results in the supplementary material. Furthermore, we are committed to making our code, data, and trained models publicly available upon publication. This will allow the research community to verify our results and build upon our work.

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

## A    MORE IMPLEMENTATION DETAILS

In the workflow optimization process of our method, we evaluate each workflow for 3 times as external execution feedback. We use the following 6 operators based on Aflow (Zhang et al., 2024b) as our base operators:

- Code Generator: generates code solutions for a given problem.
- Format Generator: produces formatted answers for a given problem.
- Ensemble Operator: combines multiple solutions or approaches to create a more robust final result.
- Review Operator: evaluates solutions for correctness, efficiency, and adherence to requirements.
- Revise Operator: refines solutions based on feedback from the review process.
- Code Test Operator: executes and validates code solutions against test cases to ensure functionality.

For our supervised fine-tuning (SFT), we utilized approximately 1,400 items sourced from three datasets generated by R1 through our optimization process. We conducted the training using LLaMA-Factory (Zheng et al., 2024), with a per-device training batch size of 1, gradient accumulation over 2 steps, a learning rate of 1e-5, and max train epochs of 3. For the reinforcement learning (RL) phase, we set the scaling factor $k$ to 1.1, the threshold $T$ to 3, rollout number $m$ to 5, and max episodes to 5. All experiments are conducted on NVIDIA A100 GPUs with 80GB of memory. The total cost of training stage is about 150 GPU hours.

The multi-purpose reward in RL is designed to balance correctness, efficiency, and exploration. However, both diversity and complexity rewards showed minimal improvement in accuracy-driven benchmarks. Consequently, we set the coefficients for these two rewards to 0.

## B    MORE CASES OF WORKFLOW

In this section, we present additional cases of both successful and failed results generated by FLOWREASONER.

### B.1    SUCCESSFUL CASES

Fig. 6, Fig. 8 and Fig. 7 demonstrate three successful cases generated by FLOWREASONER for BigCodeBench, HumanEval and MBPP.

### B.2    FAILURE CASES

Fig. 10, Fig. 11 and Fig. 9 demonstrate three failure cases generated by FLOWREASONER for BigCodeBench, HumanEval and MBPP.

## C    LIMITATION

Despite demonstrating superior performance over existing approaches, FLOWREASONER faces the limitation of the need for training, which needs some computational cost. Besides, while some generalization capability is shown when switching worker agent, performance still varies based on the worker agent used, suggesting partial dependency of the trained meta-agent on the underlying execution agent.

## D    LLM USAGE

We used an OpenAI LLM (GPT-5) as a writing and formatting assistant. In particular, it helped refine grammar and phrasing, improve clarity, and suggest edits to figure/table captions and layout (e.g., column alignment, caption length, placement). The LLM did not contribute to research ideation, experimental design, implementation, data analysis, or technical content beyond surface-level edits. All outputs were reviewed and edited by the authors, who take full responsibility for the final text and visuals.

```python
async def __call__(self, problem: str, test_cases: list):
    # Extract key problem requirements and constraints
    problem_analysis = await self.custom(
        input=problem,
        instruction=prompt_custom.GRAPH_ANALYZE_PROMPT
    )

    # Generate initial solution with problem analysis context
    solution = await self.custom_code_generate(
        problem=problem,
        entry_point=entry_point,
        instruction=f"{prompt_custom.GRAPH_GENERATE_PROMPT}\nProblem Analysis: {
            problem_analysis['response']}"
    )

    current_solution = solution['response']

    # Loop for max_iterations to improve the solution if tests fail
    for iteration in range(3):
        # Test current solution
        test_results = await self.custom(
            input=f"Solution: {current_solution}\nTest Cases: {test_cases}",
            instruction=prompt_custom.GRAPH_TEST_PROMPT
        )

        # If tests pass, return the solution
        if "failed" not in test_results['response'].lower():
            return current_solution, self.llm.cost_manager.total_cost

        # Improve solution with test feedback and problem analysis
        improved_solution = await self.custom_code_generate(
            problem=problem,
            entry_point=entry_point,
            instruction=f"{prompt_custom.GRAPH_IMPROVE_PROMPT}\nProblem Analysis: {
                problem_analysis['response']}\nCurrent Solution: {current_solution}\nTest
                Results: {test_results['response']}\nIteration: {iteration + 1}/{
                max_iterations}"
        )

        # Update current solution
        current_solution = improved_solution['response']

    # Return the last solution attempt
    return current_solution, self.llm.cost_manager.total_cost
```

Figure 6: Successful workflow generated by FLOWREASONER-14B for the BigCodeBench task of *Generating and plot weather data for a specified date range.*

```python
async def __call__(self, problem: str, test_cases: list):
    # Generate solution
    solution = await self.custom_code_generate(problem=problem, entry_point=entry_point,
        instruction=prompt_custom.GRAPH_GENERATE_PROMPT)

    # Test solution
    test_results = await self.custom(
        input=f"Solution: {solution['response']}\nTest Cases: {test_cases}",
        instruction=prompt_custom.GRAPH_TEST_PROMPT
    )

    # Optimize solution if tests fail
    if "failed" in test_results['response'].lower():
        improved_solution = await self.custom_code_generate(
            problem=problem,
            instruction=f"{prompt_custom.GRAPH_IMPROVE_PROMPT}\n\nTest Results: {test_results
                ['response']}"
        )
        return improved_solution['response'], self.llm.cost_manager.total_cost

    return solution['response'], self.llm.cost_manager.total_cost
```

Figure 7: Successful workflow generated by FLOWREASONER-14B for the MBPP task of *Writing a function to reverse words in a given string.*

```python
async def __call__(self, problem: str, test_cases: list):
    # 1. Prepare generation prompts
    gen_prompts = [
        prompt_custom.CODE_GENERATE_PROMPT_A,
        prompt_custom.CODE_GENERATE_PROMPT_B,
        prompt_custom.CODE_GENERATE_PROMPT_C,
    ]

    # 2. Generate three candidate solutions
    sols = [
        (await self.custom_code_generate(
            problem=problem,
            instruction=inst
        ))['response'] for inst in gen_prompts
    ]

    # 3. Ensemble: pick or merge the best solution
    ensemble_input = "Solutions:\n" + "\n---\n".join(sols)

    best = (await self.custom(
        input=ensemble_input,
        instruction=prompt_custom.ENSEMBLE_PROMPT
    ))['response']

    # 4. Return the best solution and total cost
    return best, self.llm.cost_manager.total_cost
```

Figure 8: Successful workflow generated by FLOWREASONER-14B for the HumanEval task of *Returning list of prime factors of given integer in the order from smallest to largest.*

```python
async def __call__(self, problem: str, test_cases: list):
    # 1. Analyze the problem
    analysis = await self.custom(
        input=problem,
        instruction=prompt_custom.ANALYZE_PROMPT
    )

    # 2. Generate initial solution
    solution = await self.custom_code_generate(
        problem=problem,
        instruction=prompt_custom.GENERATE_PROMPT,
        context=analysis['response']
    )

    # 3. Test with a subset of test cases
    sample_tests = test_cases[:min(3, len(test_cases))]
    test_result = await self.custom(
        input=f"Solution:\n{solution['response']}\nTests:\n{sample_tests}",
        instruction=prompt_custom.TEST_PROMPT
    )

    # 4. Handle based on test results
    if "all passed" in test_result['response'].lower():
        # Optimize if all tests pass
        final = await self.custom_code_generate(
            problem=problem,
            instruction=prompt_custom.OPTIMIZE_PROMPT,
            context=f"Solution:\n{solution['response']}"
        )
        return final['response'], self.llm.cost_manager.total_cost
    else:
        # Fix issues if tests fail
        fixed = await self.custom_code_generate(
            problem=problem,
            instruction=prompt_custom.FIX_PROMPT,
            context=f"Solution:\n{solution['response']}\nTest results:\n{test_result['response']}"
        )
        return fixed['response'], self.llm.cost_manager.total_cost
```

Figure 9: Failed workflow generated by FLOWREASONER-14B for the MBPP task of *Writing a function to count those characters which have vowels as their neighbors in the given string.*

```python
async def __call__(self, problem: str, test_cases: list):
    async def __call__(self, problem: str, test_cases: list):
    # Extract key problem requirements and constraints with detailed specifications
    problem_analysis = await self.custom(
        input=problem,
        instruction=prompt_custom.GRAPH_ANALYZE_PROMPT
    )

    # Initial approach planning with appropriate algorithm selection
    algorithm_selection = await self.custom(
        input=f"Problem: {problem}\nAnalysis: {problem_analysis['response']}",
        instruction=prompt_custom.GRAPH_ALGORITHM_SELECTION_PROMPT
    )

    # Generate solution with problem analysis and algorithm selection
    solution = await self.custom_code_generate(
        problem=problem,
        entry_point=entry_point,
        instruction=f"{prompt_custom.GRAPH_GENERATE_PROMPT}\nProblem Analysis: {
            problem_analysis['response']}\nAlgorithm Selection: {algorithm_selection['
            response']}"
    )

    # Self-review for logical errors and optimization opportunities
    code_review = await self.custom(
        input=f"Code: {solution['response']}",
        instruction=prompt_custom.GRAPH_CODE_REVIEW_PROMPT
    )

    # Generate improved solution with comprehensive context
    improved_solution = await self.custom_code_generate(
        problem=problem,
        entry_point=entry_point,
        instruction=f"{prompt_custom.GRAPH_IMPROVE_PROMPT}\nProblem Analysis: {
            problem_analysis['response']}\nAlgorithm Selection: {algorithm_selection['
            response']}\nCode Review: {code_review['response']}"
    )

    # Add explanation to the solution
    solution_explanation = await self.custom(
        input=f"Solution: {improved_solution['response']}\nProblem: {problem}",
        instruction=prompt_custom.GRAPH_SOLUTION_EXPLANATION_PROMPT
    )

    final_solution = f"# Solution\n{improved_solution['response']}\n\n# Explanation\n{
        solution_explanation['response']}"
    return final_solution, self.llm.cost_manager.total_cost
```

Figure 10: Failed workflow generated by FLOWREASONER-14B for the BigCodeBench task of *Extracting the text and href attributes of all anchor tags from a given URL's HTML content.*

```python
async def __call__(self, problem: str, test_cases: list):
    # 1. Generate solution
    solution = (await self.custom_code_generate(
        problem=problem,
        instruction=prompt_custom.CODE_GENERATE_PROMPT
    ))['response']

    # 2. Test and refine solution
    refined_solution = (await self.custom_code_generate(
        problem=problem,
        instruction=prompt_custom.REFINE_PROMPT,
        context=f"Initial solution:\n{solution}\n\nTest cases:\n{test_cases}"
    ))['response']

    # 3. Return the solution and total cost
    return refined_solution, self.llm.cost_manager.total_cost
```

Figure 11: Failed workflow generated by FLOWREASONER-14B for the HumanEval task of *Evaluating whether the given number n can be written as the sum of exactly 4 positive even numbers.*

