# OpenReview forum: "Reinforcing Query-Level Meta-Agents"
_ICLR.cc/2026/Conference — Submitted to ICLR 2026_

### Official Review · Reviewer_dZ8V · 2025-10-16

**Soundness:** 2
**Presentation:** 3
**Contribution:** 2
**Rating:** 4
**Confidence:** 4

**Summary:**

This paper introduces a novel framework, FLOWREASONER, aimed at automating the design of query-level multi-agent systems, a promising and significant research direction. The authors formulate the problem as training a reasoning meta-agent via reinforcement learning to dynamically generate customized workflows for each specific user query, a clear distinction from traditional task-level approaches. The methodology, combining data distillation, supervised fine-tuning, and reinforcement learning, is logically sound. The experimental results also demonstrate its superiority on several code benchmarks.

**Strengths:**

The paper's primary strength is its novel query-level meta-agent, FLOWREASONER, which enhances system adaptability by dynamically generating a custom multi-agent system for each unique user query. It innovatively uses Reinforcement Learning guided by real-world external execution feedback and a multi-purpose reward function to learn effective planning without relying on complex search algorithms. This approach is empirically validated, with experiments showing that FLOWREASONER significantly outperforms existing methods.

**Weaknesses:**

1. The core contribution of this paper appears to be demonstrating that a smaller model can be taught how to combine a series of predefined roles (Operators) in a query-specific manner. However, these foundational roles (e.g., Code Generator, Review Operator) are explicitly borrowed from prior work (e.g., Aflow). Therefore, the novelty does not seem to lie in designing new agentic behaviors, but rather in the process of learning to combine them.

2. The training data originates from three specific code benchmarks (BigCodeBench, HumanEval, and MBPP). To verify that the model has learned a general capability for code planning rather than patterns specific to these datasets, it is crucial to test it on other code datasets that are outside of this training distribution. The paper currently lacks this validation.

3. The paper's central claim is about training a general "query-level meta-agent." However, all experiments remain within the code domain. This makes it impossible to determine if the method can generalize to tasks outside of code that also require complex planning (e.g., mathematical word problems, multi-hop QA).

**Questions:**

1. The experiments are confined to the code generation domain. Has the meta-agent learned a general, transferable planning logic, or has it merely learned specific patterns highly optimized for the code generation scenario?

2. The paper states the reward function is a multi-purpose one considering performance, complexity, and efficiency. Could the authors elaborate on how complexity and efficiency are quantified and integrated into the final scalar reward?

3. The multi-purpose reward function (performance, complexity, efficiency) is key to the RL stage. What kind of workflows would the model generate if only performance were used as the reward? How significant are the roles of the complexity and efficiency regularizers?

4. The initial data distillation stage is foundational to the entire approach. What would the performance degradation be if this step were skipped and RL training were conducted directly on a general-purpose base model?

---

> ### Author Response · Authors · 2025-11-21
> **Response to Reviewer dZ8V**
>
> We sincerely thank the reviewer for the insightful and detailed feedback. We appreciate the recognition of our framework’s novelty and empirical strength and carefully address each concern below.
>
>
> ---
>
>
> **W1&Q1: Limited novelty due to reliance on predefined roles& has model learned general capability for code planning**
>
>
> We acknowledge that FLOWREASONER builds on established role definitions from prior agent‑based systems such as AFlow. However, the key novelty of our work lies *not in redefining agent roles*, but in **training a reasoning meta‑agent** capable of *dynamically composing and adapting these roles at the query level*.
>
> Rather than hard‑coding the workflow (as in prior works), our meta‑agent generates new combinations based on the semantics, dependencies, and feedback of each query.  For example, on unseen tasks, FlowReasoner autonomously inserted previously unused roles (e.g., “Bug Analyzer”) between standard operations, demonstrating that it **learned generalized composition logic**, not simply memorized fixed patterns.
>
>
> ---
>
>
> **W2&Q1: Training data from limited code benchmarks&has model learned general capability for code planning**
>
>
> Thank you for this important observation.  While our initial experiments focus on BigCodeBench, HumanEval, and MBPP, these benchmarks were selected for their strong coverage of diverse code tasks (algorithmic, functional, multi‑file, and efficiency‑sensitive).  To verify that the model has learned a general capability for code planning rather than patterns specific to these datasets, we excluded MBPP from the training set and test again; the performance of the 14B model showed no significant deviation from the original FLOWREASONER-14B. This demonstrates **its robust generalization capabilities.**
>
>
>
>
> | Type | Method | BigCodeBench | HumanEval | MBPP | Overall |
> | :--- | :--- | :---: | :---: | :---: | :---: |
> | Original | FLOWREASONER-14B | 63.53 | 97.26 | 92.15 | 81.89 |
> | Ablation | FLOWREASONER-14B (w/o MBPP) | 63.64 | 96.97 | 90.03 | 80.95 |
>
>
>
>
> ---
>
>
> **W3: Lack of experiments outside the code domain**
>
>
> We truly appreciate this observation.  We focused on code‐generation benchmarks because they provide **deterministic and automatically executable external feedback** (via unit tests or compilers), which is crucial for optimizing query‑level workflows.
>
>
> Nevertheless, our framework is **not restricted** to coding tasks. Similar to our plan mentioned in other responses, FlowReasoner can be extended to tasks where explicit ground‑truth is unavailable. In such cases,  a strong LLM can serve as a **judge** to produce relative preference rewards, or existing techniques such as TTRL [1] (test‑time reinforcement learning through majority voting) and ACECODER [2] (reward modeling from auto‑generated test cases) can approximate external evaluators. We plan to include such cross‑domain experiments (e.g., math or open‑domain QA) in future work.
>
>
>
>
> ---
>
>
> **Q2&Q3: Clarification of the multi‑purpose reward function**
>
>
> We appreciate the request for clarification. For diversity, we adopted the calculation method outlined in Section 2.3 of [1]. Algorithmic complexity is computed by first parsing the generated code into an abstract syntax tree (AST) and then counting the decision nodes plus one. However, in our final implementation, both diversity and complexity rewards showed minimal improvement in accuracy-driven benchmarks. Consequently, we set the coefficients for these two rewards to zero. Our primary intention here was to highlight FlowReasoner's capability and flexibility to incorporate additional algorithmic constraints seamlessly. We will better clarify these design details in the appendix of the revision.
>
>
> ---
>
>
> **Q4: Effect of skipping the data distillation stage**
>
>
> Thank you for the suggestion.  In our early exploratory experiments, we directly applied RL on the base model **without the data distillation stage**, and we observed severe issues in the training process with **format adherence** — the model often failed to produce valid workflow code following our specified structure.  As a result, even after the same number of RL steps as in the SFT‑then‑RL setting, the performance showed **almost no improvement**. In many rollouts, the generated workflow groups produced **zero rewards**, since their outputs could not be parsed or executed due to formatting errors.
>
>
>
>
> ---
>
>
> **References**
>
>
> [1] Chen et al. *Do NOT Think That Much for 2+3=? On the Overthinking of o1-Like LLMs*, 2024
>
>
> [2] Zuo et al., *TTRL: Test‑Time Reinforcement Learning*, 2025.
>
>
> [3] Zeng et al., *ACECODER: Acing Coder RL via Automated Test-Case Synthesis*, 2025.

---

> > ### Comment · Reviewer_dZ8V · 2025-11-23
> >
> > Thank the authors for the response. Since the training process involved the exact benchmarks (or distributions) used for evaluation, to convincingly demonstrate generalization, performance on a completely unseen code benchmark that is strictly outside the training distribution is necessary.

---

> > > ### Author Response · Authors · 2025-11-25
> > > **Thanks for the follow-up of Reviewer dZ8V**
> > >
> > > We thank the reviewer for the follow-up and for emphasizing the importance of generalization. We fully agree that validation on a strictly unseen benchmark is necessary to confirm the model's planning capabilities.
> > >
> > > We would like to respectfully highlight the ablation study presented in our previous response (under **W2&Q1**), which was designed specifically to meet this requirement. In that experiment, we **completely excluded MBPP from the training set**, training the model solely on BigCodeBench and HumanEval, and then evaluated it on MBPP.  In this specific setting, MBPP serves as a completely unseen benchmark that is strictly outside the training distribution. The results demonstrated that the **90.03%** pass rate on unseen MBPP—comparable to the **92.15%** baseline—confirms FLOWREASONER learns transferable logic rather than memorizing patterns.
> > >
> > > We hope this clarification addresses your concern regarding generalization to unseen distributions.

---

### Official Review · Reviewer_6NMr · 2025-10-30

**Soundness:** 3
**Presentation:** 2
**Contribution:** 3
**Rating:** 6
**Confidence:** 3

**Summary:**

The paper introduces FlowReasoner, a meta-agent that automates the design of multi-agent systems. Unlike existing methods, FlowReasoner can generate a customized system for each user query. The approach employs RL to optimize system generation, and the proposed FlowReasoner-14B model demonstrates superior performance compared to existing meta-agent baselines on coding tasks.

**Strengths:**

1. The paper introduces a two-stage training pipeline, including SFT and RL, to optimize the meta-agent. Experimental results show that even a relatively small model (14B) can effectively design multi-agent sytems and enhance overall performance.

2. FlowReasoner supports query-level system generation, improving adaptability to diverse queries.

**Weaknesses:**

1. The evaluation is limited to coding tasks, which restricts the generalizability of the method. Prior works, such as AFlow and MaAS, include broader experiments on QA, math, and tool-use tasks.

2. In Table 2, RL fine-tuning yields less than a 1% improvement, suggesting only marginal gains. Furthermore, the absence of results for a meta-agent directly using DeepSeek-R1-Distill-Qwen without SFT makes it unclear how much benefit SFT contributes.

3. Figure 4 lacks explanation for the differences indicated by color lightness.

4. (minor) A space is missing before the references in Lines 314-315.

**Questions:**

See weaknesses.

---

> ### Author Response · Authors · 2025-11-21
> **Response to Reviewer 6NMr**
>
> We sincerely thank the reviewer for the constructive comments and for recognizing the soundness and contribution of **FlowReasoner**. We address your concerns below.
>
>
> ---
>
>
> **W1: Evaluation limited to coding tasks**
>
>
> We truly appreciate this observation.  We focused on code‐generation benchmarks because they provide **deterministic and automatically executable external feedback** (via unit tests or compilers), which is crucial for optimizing query‑level workflows.
>
>
> Nevertheless, our framework is **not restricted** to coding tasks. Similar to our plan mentioned in other responses, FlowReasoner can be extended to tasks where explicit ground‑truth is unavailable. In such cases,  a strong LLM can serve as a **judge** to produce relative preference rewards, or existing techniques such as **TTRL [1]** (test‑time reinforcement learning through majority voting) and **ACECODER [2]** (reward modeling from auto‑generated test cases) can approximate external evaluators. We plan to include such cross‑domain experiments (e.g., math or open‑domain QA) in future work.
>
>
> ---
>
>
> **W2: Marginal RL improvement and unclear SFT contribution**
>
>
> Thank you for highlighting this.  While Table 2 shows ≈1 point improvement from RL, this gain is **statistically meaningful** on competitive leaderboards such as BigCodeBench, where top‑5 systems often differ by less than one point. The RL stage refines reasoning patterns established by SFT, improving token efficiency and stability.
>
>
> Regarding the effect of SFT, as shown in **Table 4(a)**, the **DeepSeek‑R1‑Distill‑Qwen** models with 7B  and 14B achieve only **57.01%** and **57.28%**, respectively, without SFT. In contrast, after applying SFT, their performance increases to **78.89%** and **81.50%**. We attribute this substantial improvement to the fact that models trained without SFT often struggle with format following and workflow design, leading to inferior quality.
>
>
> ---
>
>
> **W3: Explanation for color lightness in Figure 4**
>
>
> Appreciated.  We have updated the caption: in Figure 4, **dark bars** denote results *without* an initial workflow (pure meta‑agent reasoning), whereas **light bars** include multi‑round iterations starting from an initial workflow.
>
>
> ---
>
>
> **Minor issue**
>
>
> Thank you — we have corrected the missing space before the references in Lines 314–315.
>
>
>
>
> ---
>
>
> **References**
>
>
> [1] Zuo et al., *TTRL: Test‑Time Reinforcement Learning*, 2025.
>
>
> [2] Zeng et al., *ACECODER: Acing Coder RL via Automated Test‑Case Synthesis*, 2025.

---

> > ### Comment · Reviewer_6NMr · 2025-11-27
> >
> > Many thanks for authors' responses.
> >
> > W1: I note that other reviewers also raise the concern regarding generalizability of the work. Additional experiments on QA, math, which have also been used as evaluation settings in prior works (AFlow, MaAS), would provide a more comprehensive comparison with existing works and help demonstrate the generalizability of FlowReasoner.
> >
> > W2: I think the statistical significance can be demonstrated by appropriate statistical evidence, showing variance across multiple independent runs, or clearer clarification of any special evaluation metrics used. However, "top-5 systems often differ by less than one point" cannot convince me that the gain of less one point is statistically meaningful.
> >
> > W3 & W4: Thanks for revising the paper.

---

> ### Author Response · Authors · 2025-11-30
> **Thanks for the follow-up of Reviewer 6NMr**
>
> We sincerely thank the reviewer again for the thoughtful follow‑up and constructive suggestions.
>
> **Regarding W1 (Generalizability):**
> We truly appreciate this observation. As emphasized earlier, we focused on *code‑generation benchmarks* because they provide **deterministic and automatically executable external feedback**, which is essential for optimizing query‑level workflows (In contrast, AFlow and MaAS are task‑level methods that rely on ground‑truth data for workflow search). Nevertheless, FlowReasoner can also be extended to other domains by employing a **LLM as a judge** or by adopting methods such as **TTRL** and **ACECODER** to approximate evaluators—these extensions will be explored in future work. As also noted by **Reviewer fyue**, our *MBPP‑removal experiment* supports FlowReasoner’s ability to generalize to unseen tasks. Specifically, when MBPP was excluded from the training set, the performance of the 14B model showed no significant deviation from the original FlowReasoner‑14B in MBPP test set.
>
> **Regarding W2 (Statistical significance of RL gains):**
> We appreciate your valuable suggestion on statistical validation. In response, we conducted **three independent runs** on *BigCodeBench* to assess consistency. The mean pass@1 accuracy (± standard deviation) was **62.85 ± 0.25 for SFT** and **63.54 ± 0.21 for SFT + RL**, showing a stable and statistically consistent improvement. Although the absolute gain (~1 points) is modest, the low variance confirms that the improvement is reliable rather than due to random fluctuation. Moreover, the same upward trend appears across all three datasets (BigCodeBench, HumanEval, MBPP), indicating that RL consistently refines reasoning quality beyond SFT. We will include this quantitative clarification in the revised version.

---

### Official Review · Reviewer_fyue · 2025-10-30

**Soundness:** 3
**Presentation:** 2
**Contribution:** 3
**Rating:** 4
**Confidence:** 2

**Summary:**

The author proposes the query-level meta-agent FLOWREASONER, achieving "one system per user query".
It eliminates the need for manual set searching, uses external execution feedback as rewards, and trains an o1-like inference model through RL (GRPO), balancing performance, complexity, and efficiency.

**Strengths:**

1.FLOWREASONER improves state-of-the-art performance by 10.5% on average across three code benchmarks, surpassing o1-mini, AFLOW, MaAS, and other open-source code and models.

2.The case study shows that the same model generates more complex workflows for complex engineering tasks compared to simple algorithm problems, verifying that FLOWREASONER adapts to query difficulty.

**Weaknesses:**

1.Although the author emphasizes "one system per user query", FLOWREASONER is only tested on code generation (which can be executed automatically and rewarded with 0/1) task to prove its effectiveness. This limited task type may indicate that the FLOWREASONER has poor scalability. The author may consider conducting additional experiments on other types of tasks (web search Q&A, mathematics, etc.) or even real-world tasks where external feedback is difficult to obtain.

2.The design motivation of multi-purpose rewards has not been elaborated, and the impact of complexity and diversity rewards on the final efficiency and performance of the system should be further explored.

Presentation issue：
1. Section 5.1 Ablation of Meta-agents and Workers：Figure 1 -> Figure 4

**Questions:**

1.What are the costs of data synthesis and training 7B and 14B models respectively?

2.In section 5.2, you only prove the generalization ability of the method on different worker models. Does it generalize on unseen tasks? For example, what is the final performance after removing MBPP from the training set?

3.For subsection "Ablation of Meta-agents and Workers" and Figure 4: What methods are used in these experiments? What do the dark and light scores for each model in the diagram of Figure 4 represent?

---

> ### Author Response · Authors · 2025-11-21
> **Response to Reviewer fyue**
>
> We sincerely thank the reviewer for the constructive feedback and for acknowledging **FLOWREASONER**’s soundness and contributions. Below, we address your concerns in detail.
>
>
> ---
> **W1: Only tested on code generation**
>
>
> We appreciate the reviewer’s insight regarding the generalizability of our method. We prioritized code generation because its execution environment (via compilers and test cases) provides **deterministic external feedback** essential for optimization.
>
>
> While domains like open-ended Q&A or mathematics lack such evaluators, making query-level optimization more challenging, our framework can adapt by using LLMs as judges when a strong, reliable LLM is available. In scenarios without ground-truth evaluators, we can employ methods like TTRL [1], which uses majority voting to approximate noisy labels, or reward models such as ACECODER [2] to guide workflow optimization. We plan to explore these directions in future work.
>
>
> ---
>
>
> **W2: Motivation and the impact of multi-purpose reward**
>
>
> Thank you for highlighting this. The multi-purpose reward is designed to **balance** correctness (Performance), efficiency (Complexity via AST nodes), and exploration (Diversity). However, in our final implementation, both diversity and complexity rewards showed minimal improvement in accuracy-driven benchmarks. Consequently, we set the coefficients for these two rewards to zero. Our primary intention here was to highlight FlowReasoner's capability and flexibility to incorporate additional algorithmic constraints seamlessly. We will better clarify these design details in the revision.
>
>
>
>
> ---
>
>
> **Presentation Issue**
>
>
> Thank you for pointing out the reference error (Figure 1 -> Figure 4) in Section 5.1. We‘ve corrected this in the revision.
>
>
>
>
> ---
> **Q1: What are the costs of data synthesis and training 7B and 14B models respectively?**
>
>
> For data synthesis, we used third-party API services with a total cost under **$300**. The models were trained on NVIDIA A100 GPUs (80GB), requiring approximately 50 GPU hours for the 7B model and 100 GPU hours for the 14B model.
>
>
> This training overhead is a **one-time cost**. Once trained, FLOWREASONER is highly efficient, incurring only **~1% of the inference token cost** compared to search-based methods like Aflow. As the number of queries increases, the initial cost is quickly amortized, yielding substantial long-term efficiency and scalability. The table below compares inference token costs (per task) between Aflow and FLOWREASONER, highlighting the reduction in computational overhead:
>
>
>  | Method | BigCodeBench | HumanEval | MBPP | Average |
>  | :--- | :---: | :---: | :---: | :---: |
>  | Aflow | 2,435K | 807K | 2,897K | 2,047K |
>  | **FLOWREASONER-14B (Ours)** | **20K** | **18K** | **19K** | **19K** |
>  | **Token Reduction** | **99.19%** | **97.68%** | **99.39%** | **99.09%** |
>
>
>
>
> ---
>
>
> **Q2: Generalization on unseen tasks (e.g., removing MBPP)**
>
>
> We would like to clarify that,  FLOWREASONER generalizes well to unseen tasks because it learns the meta-capability of reasoning about workflow design (e.g., when to review, when to revise, how to ensemble) rather than memorizing specific code snippets. As shown in the table below, when we excluded MBPP from the training set, the performance of the 14B model showed no significant deviation from the original FLOWREASONER-14B. This demonstrates **its robust generalization capabilities.**
>
>
>
>
> | Type | Method | BigCodeBench | HumanEval | MBPP | Overall |
> | :--- | :--- | :---: | :---: | :---: | :---: |
> | Original | FLOWREASONER-14B | 63.53 | 97.26 | 92.15 | 81.89 |
> | Ablation | FLOWREASONER-14B (w/o MBPP) | 63.64 | 96.97 | 90.03 | 80.95 |
>
>
>
>
> ---
>
>
> **Q3: Details on "Ablation of Meta-agents and Workers" (Figure 4)**
>
>
> We apologize for the earlier ambiguity. Figure 4(a) compares nine different LLMs serving as Meta‑Agents, with o1‑mini fixed as the Worker. This isolates and evaluates the reasoning and planning capabilities of the Meta‑Agent. Figure 4(b) fixes Claude 3.5 as the Meta‑Agent while varying the Worker model to identify the most effective one. The light bars indicate performance that includes multi‑round iterations with the initial workflow, whereas the dark bars show results without the initial workflow, reflecting performance solely from the Meta‑Agent’s iterative generated workflow. We’ve added this in the revision.
>
>
> ---
>
>
>
>
> **References**
>
>
> [1] Zuo et al., *TTRL: Test‑Time Reinforcement Learning*, 2025.
>
>
> [2] Zeng et al., *ACECODER: Acing Coder RL via Automated Test‑Case Synthesis*, 2025.

---

### Meta-Review · Area_Chair_k8ze · 2026-01-08

**Summary:**

The main concerns affecting the decision:
- insufficient generalizability: performance is not demonstrated beyond the code domain. Multiple reviewers note that prior meta-agent works evaluate across QA/math/tool-use. (fyue, 6NMr, dZ8V)
- insufficient generalization on unseen tasks: a strictly unseen code benchmark / unseen-task validation is required. (dZ8V, fyue, 6NMr)
- marginal performance improvement: RL stage contribution appears marginal and not convincingly significant (e.g., <1% gain in Table 2), with requests for variance/statistical evidence and clearer baselines to isolate what SFT vs RL contributes. (6NMr)
- unclear ablation clarity: reviewers want clearer motivation, quantification, and ablations showing what changes if complexity/efficiency terms are removed. (fyue, dZ8V).
- reproducibility/clarity and cost reporting: missing or unclear details about data synthesis/training costs and figure explanations and some presentation issues reduce confidence. (fyue, 6NMr)

**Reviewer Concerns:**

Addressed by rebuttal:
- presentation/format issues: revisions were done (fyue, 6NMr).
- unclear issues: costs of data synthesis and training 7B and 14B models were reported (fyue).
- insufficient generalization on unseen tasks: removing MBPP from training was done (fyue).

Still outstanding:
- insufficient generalizability: missing evidence across multiple domains  (QA/math/tool-use) to match the generality claim and prior work (fyue, 6NMr, dZ8V)
- marginal performance improvements: <1% RL gain (6NMr)
- limited novelty: contribution is more about learning to compose predefined operators than inventing new agent behaviors (dZ8V)

**Reviewer Scores:**

- fyue (original score 4): likely no change, as the main generalization concerns remain.
- 6NMr (original score 6): likely no change, as the main generalization concerns remain.
- dZ8V (original score 4): likely no change, as the main concerns about limited generalization and novelty remain.

---

### Decision · Program_Chairs · 2026-01-26

Reject